# GHOST: Unmasking Phantom States in `Mamba2` via Grouped Hidden-state Output-aware Selection & Truncation

Michael Menezes [1]   Anastasios Kyrillidis [1]

## Abstract

While `Mamba2`'s expanded state dimension enhances temporal modeling, it incurs substantial inference overhead that saturates bandwidth during autoregressive generation. Standard pruning methods fail to address this bottleneck: unstructured sparsity leaves activations dense, magnitude-based selection ignores runtime dynamics, and gradient-based methods impose prohibitive costs. We introduce GHOST (Grouped Hidden-state Output-aware Selection and Truncation), a structured pruning framework that approximates control-theoretic balanced truncation using only forward-pass statistics. By jointly measuring controllability and observability, GHOST rivals the fidelity of gradient-based methods without requiring backpropagation. As a highlight, on models ranging from 130M to 2.7B parameters, our approach achieves a 50% state-dimension reduction with approximately 1 perplexity point increase on WikiText-2. Code is available at https://github.com/Menezmic21/mamba2_ghost.

## 1. Introduction

Driven by scaling laws, the transition from `Mamba1` to `Mamba2` increased the SSM state dimension $N$ from 16 to 128 (Dao & Gu, 2024a); for a 1.3B model, this swells the recurrent state from $\approx$ 12 MB to $\approx$ 100 MB. While enhancing temporal modeling, this $8\times$ expansion creates an inference bottleneck by saturating memory bandwidth and degrading cache locality (Asif et al., 2025). Effective post-training compression is essential to democratize the deployment of large-scale SSMs. However, pruning the state dimension of `Mamba2` presents challenges that expose the limitations of transformer-based compression paradigms.

First, *unstructured methods can be insufficient for inference acceleration.* While second-order optimizers like OBC (Dao & Gu, 2024c) and SparseGPT (Frantar & Alistarh, 2023), or simple activation-aware metrics like Wanda (Sun et al., 2024), effectively sparsify projection weights, their original formulations generally yield *dense activations*. Since the product of a sparse weight row and a dense input vector is non-zero, the resulting projected input remains dense. Consequently, `Mamba2`'s recurrent state $H_t$ (to be defined later in text) remains fully populated, foregoing reductions in memory bandwidth consumption.

Second, *static magnitude pruning is often blind to system dynamics.* Standard magnitude pruning (Saikumar & Varghese, 2025) evaluates a state channel based on fixed weight norms rather than its dynamic energy. This reliance on static proxies creates a "blind spot" where the pruner cannot distinguish between a state's theoretical capacity and its actual utilization in `Mamba`. We categorize the resulting failure modes into *corporeal states*, false-negative channels which appear structurally significant but are inert under dynamics, and *phantom states*, false-positive channels which exhibit low weight norms but high activity. As illustrated in Figure 1, there is often no positive correlation between a state's static magnitude and its dynamic energy. So, standard pruning inadvertently discards high-activity phantom states while preserving inert corporeal ones.

Third, *gradient-based solutions can be expensive.* While structured methods like Taylor pruning (Ghattas et al., 2025) offer high fidelity by estimating loss sensitivity, they require computing and storing gradients for the full computational graph. This consumes 45 GB VRAM for a 1.3B model that exceeds e.g., the standard 40 GB A100 capacity. In contrast, our proposed method utilizes only 15 GB. To offset prohibitive costs, gradient-based methods necessitate "one-shot" estimation, which suffers from distribution shift as upstream pruning invalidates downstream gradients (Dao & Gu, 2024c).

**This paper.** We introduce Grouped Hidden-state Output-aware Selection and Truncation, or GHOST, a structured pruning framework that approximates balanced truncation

[1]Department of Computer Science, Rice University, Houston, TX, United States. Correspondence to: Anastasios Kyrillidis <anastasios@rice.edu>.

*Proceedings of the 43rd International Conference on Machine Learning*, Seoul, South Korea. PMLR 306, 2026. Copyright 2026 by the author(s).

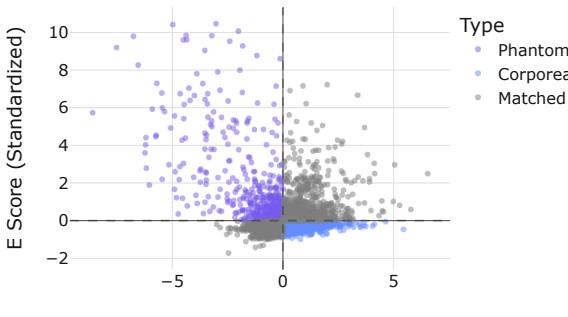

*Figure 1.* **The Proxy Failure: Static Magnitude vs. Dynamic Energy.** We correlate weight-based importance and runtime usage across `Mamba2`-1.3B's 6,144 hidden states (48 layers, 128 states/layer). Akin to an element of a Transformer's KV cache, a state is a coordinate in `Mamba2`'s recurrent memory. Because $W_B$ maps inputs to states and $W_C$ maps states to outputs, weight magnitude ($W_{\text{score}} = \sqrt{\|W_{B_i}\|_2 \|W_{C_i}\|_2}$) may seem a naive importance proxy. However, the standardized $x$-axis (static score) and $y$-axis (dynamic energy) show a weak relationship with an average Pearson Correlation of $-0.1940$. At 50% sparsity, we see magnitude-based pruning suffer a 39.2% total misclassification rate, split between **Phantom States** (20.18%; top-left) and **Corporeal States** (19.05%; bottom-right). GHOST identifies the former and prunes the latter, correcting this misalignment.

(Moore, 1981), using only forward-pass statistics. By computing the product of empirical controllability (how effectively inputs drive a channel) and observability (how strongly a channel influences outputs) from calibration data, GHOST identifies and preserves "living" states that are simultaneously reachable and impactful.

GHOST operates within `Mamba2`'s Grouped Query Attention (GQA) structure, where groups of heads share dynamics parameters. We pool saliency scores across groups within each layer, enabling adaptive capacity allocation: groups modeling complex dynamics retain more states, while redundant groups are aggressively pruned. Sequential layer-by-layer processing with activation updates mitigates distribution shift, and the entire procedure requires only two forward passes per layer.

Our main contributions are as follow:

- We propose GHOST, a data-driven structured pruning framework for `Mamba2` that utilizes inter-group thresholding to prune the SSM state dimension $N$ based on measured controllability and observability.

- We demonstrate that GHOST solves the "proxy failure" of `Magnitude` pruning, successfully exorcising *corporeal* and preserving *phantom* states.

- We provide empirical evidence that GHOST outperforms `Magnitude`, and `Random` baselines, and rivals the per-

formance of `Taylor` pruning, while requiring *no backpropagation* and a fraction of the cost.

## 2. Preliminaries

Here, we provide a brief overview of the `Mamba2` architecture. A more detailed exposition is found in Appendix A. Readers familiar with these topics may proceed to Section 3.

Given input $u_t \in \mathbb{R}^M$, the model applies RMSNorm and a linear transformation to produce the gate $z_t$, input $x_t$, dynamics parameters $B_t, C_t$, and timescale $\Delta_t$[1]:

$$[z_t; x_t; B_t; C_t; \Delta_t] = W_{\text{in}} \text{RMSNorm}(u_t) + b_{\text{in}}, \quad (1)$$

where $z_t, x_t \in \mathbb{R}^{H \times P}, \Delta_t \in \mathbb{R}^H$ are unique to each of the $H$ heads, while $B_t, C_t \in \mathbb{R}^{G \times N}$ are shared within $G$ groups of $K = H/G$ heads. We use $W_B, W_C$ to refer to the slices of $W_{\text{in}}$ responsible for generating $B_t, C_t$. Following a depthwise convolution and SiLU activation on the signal and dynamics branches (yielding $x'_t, B'_t, C'_t$), the sequence undergoes discretized recurrence using learned parameters $A, D \in \mathbb{R}^H$ with $A_h \in \mathbb{R}^+$ for all $h \in [H]$. For head $h$ belonging to group $g_h$, the hidden state $H_{t,h} \in \mathbb{R}^{N \times P}$ and output $y^{\text{SSM}}_{t,h} \in \mathbb{R}^{1 \times P}$ are updated as:

$$H_{t,h} = \overline{A}_{t,h} H_{t-1,h} + \overline{B}^{\top}_{t,h} x'_{t,h}, \quad (2)$$

$$y^{\text{SSM}}_{t,h} = C'_{t,g_h} H_{t,h} + D_h x'_{t,h}, \quad (3)$$

where $\overline{A}_{t,h} = \exp(-\Delta'_{t,h} A_h), \overline{B}_{t,h} = \Delta'_{t,h} B'_{t,g_h}$, and $\Delta'_{t,h} = \text{Softplus}(\Delta_{t,h})$. The final block output is obtained by gating $y^{\text{SSM}}_t$ with $\text{SiLU}(z_t)$, normalizing, projecting with $W_{\text{out}} \in \mathbb{R}^{M \times H \cdot P}$, and adding a residual.

## 3. Related Work

Existing SSM compression methodologies are often incompatible with `Mamba2`'s architectural innovations or orthogonal to state-dimension pruning. For instance, `SparseSSM` (Tuo & Wang, 2025) targets the diagonal $A$ matrix of `Mamba1` to determine state importance; this metric is unavailable in `Mamba2`, as its scalar-identity formulation lacks state-specific granularity. Similarly, timescale-based methods like `PerfMamba` (Asif et al., 2025) and $\Delta$-guided pruning (Anonymous, 2025) operate at the granularity of heads; while effective for head-pruning, they cannot resolve intra-head state redundancy. Coarser techniques like `Mamba-Shedder` (Muñoz et al., 2025) remove entire architectural blocks, lacking the granularity required to compress the state dimension $N$. Finally, while gradient-based structured pruning discussed in Section 1 is theoretically

---

[1]We follow the notation used in the literature, but we acknowledge that there is ambiguity about what is a vector (usually boldface, lowercase letters) and we have to make exceptions like $\Delta_t \in \mathbb{R}^H$. A complete notation table is provided in Appendix B.

applicable, it suffers from the *masked distribution shift* problem (Dao & Gu, 2024c) and prohibitive memory overheads.

# 4. Methodology

We propose GHOST (Grouped Hidden-state Output-aware Selection and Truncation), a structured pruning framework designed to compress the state dimension $N$ of `Mamba2` as Figure 2 illustrates.

## 4.1. Problem Formulation

We consider a pre-trained `Mamba2` model $f(\cdot; \boldsymbol{\theta})$, where we abstractly denote trainable parameters as $\boldsymbol{\theta}$. Our objective is to identify a binary mask $\boldsymbol{M} \in \{0,1\}^{N_{\text{layers}} \times G \times N}$ that zeroes out specific channels in the state dimension $N$. Pruning $N$ directly reduces the size of the recurrent state $\boldsymbol{H}_t$, yielding strictly lower activation memory and FLOPs during recurrence. We minimize prediction error on a calibration set $\mathcal{D}_{\text{cal}}$, subject to a target state sparsity $\kappa \in [0, 1]$:

$$\min_{\boldsymbol{M}} \quad \mathbb{E}_{x \sim \mathcal{D}_{\text{cal}}}[\mathcal{L}(f(x; \boldsymbol{\theta}|_{\boldsymbol{M}}))]$$

$$\text{s.t.} \quad \frac{\|\boldsymbol{M}\|_0}{N_{\text{layers}} \cdot G \cdot N} \leq 1 - \kappa.$$

Pruning the state dimension directly reduces the memory footprint of $\boldsymbol{H}_t$ from $(H, N, P)$ to $(H, N', P)$ where $N' = (1 - \kappa)N$, with proportional reductions in memory bandwidth during autoregressive generation.

## 4.2. Theoretical Motivation: Balanced Truncation

GHOST approximates the principles of *balanced truncation* (Moore, 1981) to identify and prune redundant state channels. Fundamentally, this approach seeks to retain states that serve as effective conduits for information flow, characterized by: (1) **Controllability:** How much does each previous input impact the current state; and (2) **Observability:** How much does the current state impact future outputs. In the context of standard state-space models, the most critical states are those that are both well used under the input history and have a significant impact on future predictions.

The projection in Equation (1) characterizes `Mamba2` as a Linear Time-*Varying* (LTV) system governed by the input-dependent terms $\boldsymbol{\Delta}'_t$, $\boldsymbol{B}'_t$, and $\boldsymbol{C}'_t$. Moreover, the SiLU activation that follows the depthwise convolution renders the system non-commutative. These properties preclude the use of standard algebraic solutions (e.g., Lyapunov equations), necessitating a time-varying, data-driven approach.

## 4.3. Empirical Estimation via Moments and Hessians

To adapt controllability and observability for the `Mamba2` architecture, we use instantaneous empirical Gramians (Lall et al., 2002; Himpe, 2018) evaluated at time $t$ over a sequence of length $L$.

**Second Hidden State Moment as Controllability.** We quantify the extent to which the input history uses a state by analyzing the statistics of the hidden states themselves. Since $\boldsymbol{H}_{t,h}$ is the accumulation of processed inputs, its magnitude represents the energy transferred from the input. Per Himpe (2018), we define the empirical measure of controllability as the *second hidden state moment*, capturing the statistical dispersion of the state induced by the data.

**Output Energy Hessian as Observability.** To quantify the influence of the current state on future outputs, we examine the curvature of the output energy with respect to the state. We define observability as the *Hessian of* $\sum_{s=t}^{L} \frac{1}{2}(\boldsymbol{y}_{s,h,p}^{SSM})^2$ *with respect to the hidden state* $\boldsymbol{H}_{t,h,n,p}$. For the immediate output $\boldsymbol{y}_{t,h,p}^{\text{SSM}} = \boldsymbol{C}'_{t,g_h} \boldsymbol{H}_{t,h,\cdot,p} + \boldsymbol{D}_h \boldsymbol{x}'_{t,h,p}$,

$$\nabla^2_{\boldsymbol{H}_{t,h,n,p}} \frac{1}{2}(\boldsymbol{y}_{t,h,p}^{\text{SSM}})^2 = (\boldsymbol{C}'_{t,g_h})^\top \boldsymbol{C}'_{t,g_h}.$$

While a full observability metric would sum these Hessians over all future time steps $k \geq t$, we adopt a local approximation based on $(\boldsymbol{C}'_{t,g_h})^\top \boldsymbol{C}'_{t,g_h}$. This is justified by the decay inherent in the discretized transition dynamics, which suggests that the contribution of future outputs to the current state's observability diminishes exponentially (Gwak et al., 2025). As evidence by Appendix C, Table 8, this instantaneous Hessian captures the dominant term of the full observability Gramian while providing an efficient proxy that avoids backward passes over the temporal horizon.

**Saliency Scoring.** Given that $\boldsymbol{A}_h$ is a scalar and the state channel recurrences in Equation (2) are independent, we restrict our analysis to the diagonal terms of controllability. Likewise for observability, as standard in Hessian-based pruning (LeCun et al., 1989), we adopt a diagonal approximation of the output energy Hessian. For a specific state channel $n \in [N]$ within the $k$-th head in group $g$ (head $h_{k,g}$ or simply $h$ for convenience), and head channel $p$, we define the instantaneous diagonal scores as:

$$\boldsymbol{P}_{k,n,p}^{(g)}(t) = (\boldsymbol{H}_{t,h,n,p})^2, \qquad \boldsymbol{Q}_n^{(g)}(t) = (\boldsymbol{C}'_{t,g,n})^2.$$

Moreover, we define the instantaneous saliency $\boldsymbol{S}_{k,n,p}^{(g)}(t) = \boldsymbol{P}_{k,n,p}^{(g)}(t) \cdot \boldsymbol{Q}_n^{(g)}(t)$ as the product of these terms, mirroring the construction of Hankel singular values in balanced truncation (Moore, 1981). To obtain a robust global estimate, we aggregate this score over the sequence length $L$ and the calibration dataset $\mathcal{D}_{\text{cal}}$. Reducing the GQA structure via expectation, the final saliency score for state channel $n$ is:

$$\boldsymbol{S}_n^{(g)} = \sqrt{\frac{1}{Z} \sum_{d=1}^{|\mathcal{D}_{\text{cal}}|} \sum_{t=1}^{L} \sum_{k=1}^{K} \sum_{p=1}^{P} S_{k,n,p}^{(g)}(t|d)}, \qquad (4)$$

where $Z = |\mathcal{D}_{\text{cal}}| \cdot K \cdot P$ is the normalization constant. The final square root aligns our metric with the theoretical defini-

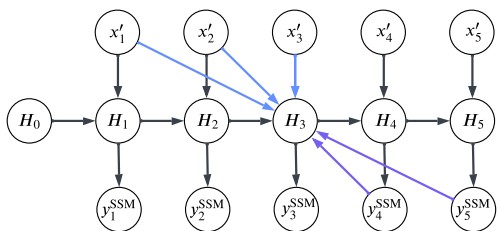
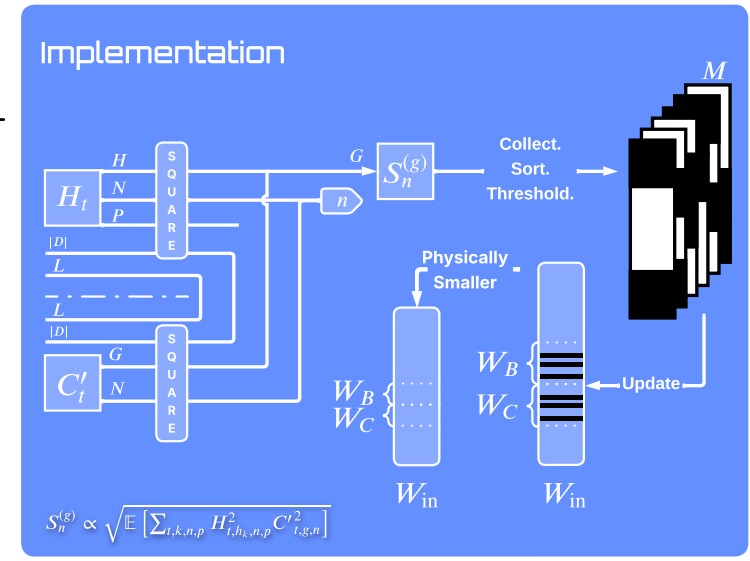

*Figure 2.* Overview of the GHOST mechanism. On the left, we formulate the importance of a hidden state channel as the product of its *controllability* (measured by channel variance from inputs $x'_t$) and *observability* (approximated by the Hessian of output energy from outputs $y_t^{\text{SSM}}$). On the right, we demonstrate how hidden states $H_t$ and variables $C'_t$ are squared and combined to compute the channel importance scores $S_n^{(g)}$. These scores are collected, sorted, and thresholded to generate a mask $M$. This mask is then applied to update the input weight matrix $W_{\text{in}}$ (specifically targeting slices $W_B$ and $W_C$), which effectively prunes the network and results in a physically smaller, sparsified weight representation. See Appendix B, Figure 6 for a more exhaustive depiction of the `Mamba2` forward pass.

tion of Hankel singular values ($\sigma_i \approx \sqrt{\lambda_i(PQ)}$). Beyond maintaining theoretical fidelity, this transformation restores the dimensionality of the score to the linear scale of the input features, facilitating direct magnitude comparisons.

### 4.4. Impact on Local Loss

Although we derived our salience score using control theory, it is consistent with a loss preservation perspective. Due to the GQA structure of `Mamba2`, pruning a state channel $n$ within group $g$ only affects the subset of $K$ heads associated with that group. Let $\mathcal{L}_t^{(g)}$ be the local reconstruction error at time $t$ for group $g$, defined as the squared Frobenius norm of the difference between the pruned and original outputs for the affected heads. Following standard analysis in SSM literature (Gu & Dao, 2024; Dao & Gu, 2024b), we set $D = 0$ as the feedthrough term does not contribute to state dynamics, the primary targets of our pruning.

$$\mathcal{L}_t^{(g)}(H'_t) = \sum_{k=1}^{K} \|y'^{\text{SSM}}_{t,h_k} - \hat{y}^{\text{SSM}}_{t,h_k}\|_F^2$$
$$= \sum_{k=1}^{K} \|C'_{t,g} H'_{t,h_k} - C'_{t,g} H_{t,h_k}\|_F^2,$$

where $h_k$ denotes the global index of the $k$-th head in group $g$. The error simplifies to the impact of the state perturbation $\delta H_{t,h_k} = H'_{t,h_k} - H_{t,h_k}$. Pruning channel $n$ corresponds to zeroing out the $n$-th row of the hidden state matrix for

these specific heads. Substituting $\delta H_{t,h_k,n} = -H_{t,h_k,n}$ yields:

$$\mathcal{L}_t^{(g)} = \sum_{k=1}^{K} \|C'_{t,g}(\delta H_{t,h_k})\|_F^2$$
$$= \sum_{k=1}^{K} \sum_{p=1}^{P} (H_{t,h_k,n,p})^2 (C'_{t,g,n})^2.$$

We seek a pruning decision robust across the data distribution. Averaging over calibration set $\mathcal{D}_{\text{cal}}$ and summing over sequence length $L$, we define the *expected cumulative error*:

$$\mathbb{E}[\mathcal{L}] = \mathbb{E}\left[\sum_{t=1}^{L} \mathcal{L}_t^{(g)}\right]$$
$$= \frac{1}{|\mathcal{D}_{\text{cal}}|} \sum_{d=1}^{|\mathcal{D}_{\text{cal}}|} \sum_{t=1}^{L} \sum_{k=1}^{K} \sum_{p=1}^{P} (H_{t|d,h_k,n,p})^2 (C'_{t|d,g,n})^2.$$

Observing that $S_n^{(g)} \propto \sqrt{\mathbb{E}[\mathcal{L}]}$, we conclude that minimizing the GHOST saliency score $S_n^{(g)}$ is equivalent to minimizing the expected increase in local mean-squared error.

### 4.5. Inter-Group Thresholding and Pruning

GHOST performs *inter-group thresholding*. We pool the scores from all $G$ groups within the current layer:

$$\mathcal{S}_{\text{pool}} = \bigcup_{g=1}^{G} \{S_n^{(g)} \mid n \in [N]\}.$$

We sort these $G \times N$ scores and determine a threshold $\tau$ based on the target sparsity for that layer. This dynamically allocates capacity: complex groups retain more channels ($r > N(1 - \kappa)$), while redundant ones are heavily pruned.

Our original formulation zeroed out projection weights $\boldsymbol{W}_B, \boldsymbol{W}_C$ and corresponding convolution filters according to the binary mask derived from $\tau$. However, to achieve actual hardware speedups, we now initialize a new layer with a reduced state dimension. State-independent parameters are copied directly, while state-dependent tensors (e.g., $\boldsymbol{W}_B, \boldsymbol{W}_C$) are sliced using the binary mask. The original layer is then replaced and deleted. The model is updated sequentially, using the modified output activations from the current layer to calibrate the next layer, ensuring subsequent layers adapt to the sparsified dynamics.

### 4.6. Complexity Analysis

GHOST, Algorithm 1, is computationally efficient, requiring only two forward passes over the calibration data. In practice, we compute raw sums rather than the root mean square shown in Equation (4), as the raw values are rank equivalent and faster to compute.

---

**Algorithm 1** GHOST: Given `Mamba2` layer `mixer` and inputs $\mathcal{X}_{\text{in}}$, we prune the SSM state dimension to $\kappa$ sparsity.

$\mathcal{S}_{\text{pool}} \leftarrow \emptyset$
$\boldsymbol{S}^{(g)} \leftarrow \boldsymbol{0}_N \; \forall g$
**for** $\boldsymbol{u} \in \mathcal{X}_{\text{in}}$ **do**
  **for** $t = 1$ **to** $L$ **do**
    $_-, \boldsymbol{H}_t, \boldsymbol{C}'_t \leftarrow \text{mixer}(\boldsymbol{u}_t)$ // transient states
    $\boldsymbol{S}_n^{(g)} \leftarrow \boldsymbol{S}_n^{(g)} + \sum_{k,p}(\boldsymbol{H}_{t,h_k,n,p})^2 \cdot (\boldsymbol{C}'_{t,g,n})^2 \; \forall g, n$
  **end for**
**end for**
$\mathcal{S}_{\text{pool}} \leftarrow \{\boldsymbol{S}^{(g)} : g \in [G]\}$ // inter-group pool
$\boldsymbol{M} \leftarrow$ mask of $\kappa \cdot G \cdot N$ states with lowest $\mathcal{S}_{\text{pool}}$ rank
$\text{mixer} \leftarrow \text{PruneMaskedStates}(\text{mixer}, \boldsymbol{M})$
$\mathcal{X}_{\text{out}, \_, \_} \leftarrow \text{mixer}(\mathcal{X}_{\text{in}})$ // update activations
**return** $\text{mixer}, \mathcal{X}_{out}$

---

**Time Complexity:** $O(|\mathcal{D}_{\text{cal}}| \cdot L \cdot G \cdot K \cdot P \cdot N)$ per layer. This matches the cost of a standard inference pass, as we accumulate squared values during computation.

**Space Complexity:** $O(G \cdot N)$. We only store $N$ statistics per group, avoiding the $O(N^2)$ storage required by full-Gramian methods or Hessian-based pruning.

## 5. Experiments

We evaluate GHOST across sparsity, sequence length, model scale, tasks, and out-of-distribution robustness. Unless otherwise noted, we fixed random seeds to 42 and used `Mamba2`-1.3B with Huggingface default precision (float32)

and sequence length (2048) (Wolf et al., 2020). We calibrated on 128 samples of WikiText-2 (Merity et al., 2016) at a batch size of one to save memory and compressed to 50% sparsity. We measured performance using EleutherAI's LMEval (Gao et al., 2024) with a H100 80 GB GPU.

Figure 3 illustrates the computational trade-offs of the evaluated methods. Regarding time efficiency (left), `Magnitude` and `Random` are essentially free. Among the data-driven methods, while all share linear complexity with respect to the pruning target $N$, the critical differentiator is their scaling with the dimension $M$. Here, `SparseGPT` scales quadratically, whereas `Taylor` and GHOST remain linear. Regarding memory constraints (right), `Taylor`'s reliance on gradient computation incurs significant overhead, exceeding the capacity of standard A100 40 GB GPUs.

In the following experiments, we group `SparseGPT` with the `Dense` baseline. Unlike structured methods that remove entire channels, unstructured pruning deletes individual weight elements; this allows `SparseGPT` to leverage the remaining weights to re-optimize and recover performance. Consequently, it serves as an *optimistic* baseline for accuracy and an *unfair* baseline for efficiency. We empirically highlight this discrepancy in Appendix C, Table 9.

### 5.1. Impact of Sparsity Levels

To assess the robustness of GHOST against varying degrees of compression, we conducted a sparsity sweep ranging from 10% to 90% sparsity. We compared the post-pruning perplexity on WikiText-2 against competing methods. The dense baseline attained a perplexity of 13.17.

The results in Table 1 demonstrate a distinct trend: while `Taylor` exhibited robust performance at lower sparsity levels, it destabilized as the pruning threshold increased. To mitigate the high computational cost of backpropagation, `Taylor` relies on one-shot score computation. However, this efficiency trade-off renders it susceptible to catastrophic collapse, as it fails to account for distribution shifts induced by pruned layers (Appendix C, Table 10 and Figure 7). In contrast, GHOST's lower cost allows sequential updates to ease this effect. Moreover, GHOST remained competitive with `Taylor` and `SparseGPT` in low-sparsity regimes. Crucially, despite operating under computational and structural constraints, GHOST maintained a smooth degradation curve, yielding a viable model even at 70% sparsity: a regime where other structural baselines exploded.

### 5.2. Sequence Length Generalization

For computing empirical Gramians, GHOST assumes a target sequence length. We investigated how this impacts sequence length generalization by calibrating all methods on short contexts, $L_{\text{cal}} = 128$, and evaluated perplexity on

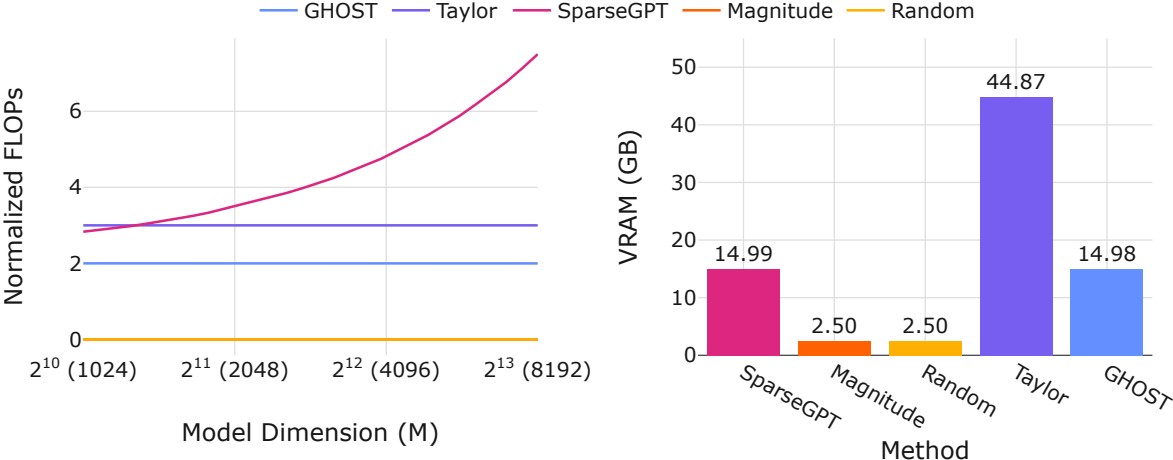

*Figure 3.* **Time and Space Efficiency.** (Left) FLOPs normalized by the computation required for a single forward pass. (Right) Peak VRAM requirements by method. Note that `Taylor` exceeds the memory capacity of 40 GB cards.

*Table 1.* **Sparsity Sweep.** Perplexity comparison on WikiText-2 across increasing sparsity levels. While top methods perform comparably at moderate sparsity (50%), GHOST demonstrates significantly better retention of model capabilities in the high-sparsity regime ($> 70\%$), avoiding the catastrophic breakdown observed in `Taylor`.

| Method | Target Sparsity | | | | |
| --- | --- | --- | --- | --- | --- |
| | **10%** | **30%** | **50%** | **70%** | **90%** |
| SparseGPT | 13.17 | 13.19 | 13.25 | 13.52 | 15.51 |
| Magnitude | 427.7 | 22.39 | 29.34 | 39.96 | 69.49 |
| Random | 13.52 | 15.04 | 17.77 | 30.14 | 64.21 |
| Taylor | **13.18** | **13.26** | **13.94** | 4255 | 4690 |
| GHOST | 13.24 | 13.51 | 14.23 | **16.16** | **25.07** |

*Table 2.* **Length Generalization.** Comparison of perplexity degradation when evaluating on sequence lengths longer than the calibration context. We denote calibration free methods with *.

| Method | $L_{\text{cal}}$ | Evaluation Length ($L_{\text{eval}}$) | | | |
| --- | --- | --- | --- | --- | --- |
| | **128** | **256** | **512** | **1024** | **2048** |
| Dense* | 23.04 | 18.52 | 15.88 | 14.20 | 13.18 |
| SparseGPT | 23.11 | 18.59 | 15.96 | 14.29 | 13.27 |
| Magnitude* | 32.76 | 27.96 | 26.38 | 27.32 | 29.34 |
| Random* | 27.81 | 22.89 | 20.17 | 18.59 | 17.77 |
| Taylor | 317.3 | 386.9 | 497.8 | 797.0 | 1613 |
| GHOST | **23.99** | **19.46** | **16.85** | **15.22** | **14.27** |

increasingly long contexts up to $L_{\text{eval}} = 2048$.

Table 2 shows that GHOST generalizes well: perplexity decreases with longer evaluation contexts, following the same trend as the dense model. In contrast, `Taylor`'s perplexity increases with sequence length, reaching 1613 at $L_{\text{eval}} = 2048$—a dramatic failure suggesting that gradient-based scores computed on short contexts do not transfer to long-range dependencies.

### 5.3. Sequence Length Robustness

Beyond sequence length generalization, we evaluated robustness on identical, progressively shorter calibration and testing lengths. As illustrated in the log-log plot in Figure 4, while most methods exhibit a gentle perplexity increase at shorter lengths, `Taylor` displays a distinct anomalous behavior. It diverges significantly for $L \in [16, 512]$, un-

derperforming even trivial baselines like `Magnitude` and `Random`, before rapidly recovering strong performance outside this window (see Appendix C, Table 12 for an interpretability study). In contrast, GHOST maintains consistent stability across the sweep, tracking the `SparseGPT` and `Dense` baselines. A tabulation is presented in Appendix C, Table 11.

### 5.4. Scaling Laws

We evaluated the efficacy of pruning across varying model capacities, ranging from 130M to 2.7B parameters. Smaller models typically exhibited lower parameter redundancy, making them significantly more sensitive to the removal of state dynamics. As shown in Table 3, this regime exposes critical instabilities in `Taylor` which failed catastrophically on the 130M and 370M models, resulting in perplexity explosions ($> 10^4$), likely due to the noisy gradient signals inherent in smaller state-space models.

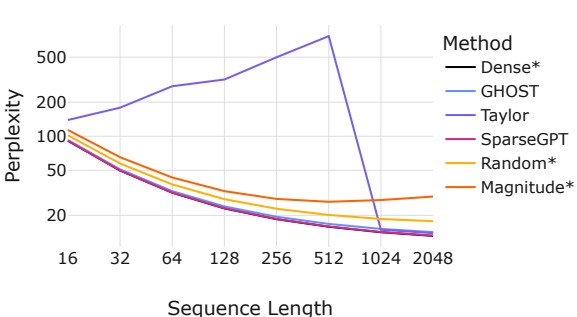

*Figure 4.* **Length Robustness.** Comparison of perplexity degradation when evaluating on increasingly shorter contexts. We denote calibration free methods with *.

*Table 3.* **Scaling Laws.** Perplexity comparison across `Mamba2` model sizes. While `Taylor` collapses on smaller architectures due to gradient instability, `GHOST` remains robust across all scales.

| Method | 130M | 370M | 780M | 1.3B | 2.7B |
|---|---|---|---|---|---|
| Dense | 25.85 | 18.13 | 14.97 | 13.17 | 11.46 |
| SparseGPT | 26.17 | 18.34 | 15.08 | 13.25 | 11.51 |
| Magnitude | 910.0 | 120.3 | 30.57 | 29.34 | 20.04 |
| Random | 45.60 | 23.71 | 5197 | 17.77 | 15.93 |
| Taylor | 1E16 | 12647 | **15.13** | **13.94** | **11.50** |
| GHOST | **29.06** | **19.96** | 16.13 | 14.23 | 12.14 |

In contrast, `GHOST` demonstrated superior stability. It outperformed all baselines on the 130M and 370M models, proving that its feature-based selection criterion effectively identifies essential dynamics even when redundancy is scarce. While `Taylor` recovered performance at larger scales (1.3B and 2.7B) where gradient signals presumably stabilize, `GHOST` remained the only method to provide consistent, usable post-pruning performance across the entire scaling spectrum.

### 5.5. Zero-Shot Performance

To analyze flexibility across data domains, we calibrated and evaluated all methods on various zero shot tasks: Lambada (Paperno et al., 2016), PIQA (Bisk et al., 2019), ARC-e, and ARC-c (Clark et al., 2018). See Figure 5.

As expected, the unpruned `Dense` model provides an upper bound for performance, while `SparseGPT` generally retains high fidelity. Among the structured pruning baselines, naive approaches such as `Magnitude` and `Random` pruning suffer catastrophic performance drops on the Lambada dataset, falling to 17.50% and 31.21% respectively. `Taylor` pruning demonstrates robustness, achieving the

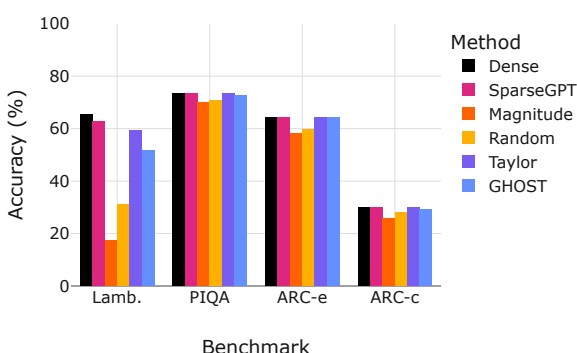

*Figure 5.* **Zero-Shot Performance.** Accuracy on Lambada, PIQA, ARC-e, and ARC-c.

highest accuracy among compressed methods on Lambada (59.23%), ARC-e (64.27%), and ARC-c (29.86%). Our proposed method, `GHOST`, remains highly competitive, even matching `Taylor` at a fraction of the cost on the PIQA dataset with an accuracy of 64.27%, which is 0.04% above the dense baseline. We use a bar chart to highlight the close trend between `Taylor` and `GHOST`. See Appendix C, Table 13 for a tabulation.

### 5.6. Out-of-Distribution (OOD) Robustness

Tensor-wise, Hessian-based pruning often overfits the curvature of the specific calibration domain (e.g., English text) (Tang & Yang, 2025). `GHOST`, acting as a structural regularizer based on system dynamics, should offer better generalization to unseen distributions. We calibrated on WikiText-2 and evaluated zero-shot performance on code generation (HumanEval) and multiple choice math problems (MMLU Elementary Mathematics). As hypothesized, `GHOST` outperforms the baselines on OOD tasks (Table 4). This suggests that `GHOST` identifies channels that are fundamentally important to the State Space Model's operation, rather than those that are merely important for predicting the next word in English prose.

### 5.7. Calibration Data Efficiency

We evaluated the perplexity of the pruned models while varying the number of calibration samples $|\mathcal{D}_{\text{cal}}| \in \{8, 16, 32, 64, 128\}$. Consistent with Ji et al. (2025), Table 5 shows all methods feature comparable sample efficiency.

### 5.8. Saliency Ablation

To evaluate the importance of the mixed controllability and observability score, we conducted a sparsity sweep ranging from 10% to 90% sparsity while changing the definition of $S_n^{(g)}(t)$ between $P_{k,n,p}^{(g)}(t)$, $Q_n^{(g)}(t)$, and $P_{k,n,p}^{(g)}(t) \cdot Q_n^{(g)}(t)$.

*Table 4.* **OOD Robustness.** Performance on OOD datasets, HumanEval and MMLU Elementary Math, after calibrating solely on WikiText-2. GHOST shows the smallest gap between in-domain (ppl) and out-of-domain (acc) performance.

| Method | In-Domain | Out-of-Domain | |
|---|---|---|---|
| | Text | Code | Math |
| Dense | 13.17 | 6.71 | 21.43 |
| SparseGPT | 13.25 | 5.49 | 22.75 |
| Magnitude | 29.34 | 0.61 | **25.66** |
| Random | 17.77 | 4.27 | 21.43 |
| Taylor | **13.94** | 4.88 | 22.49 |
| GHOST | 14.23 | **5.49** | **25.66** |

*Table 5.* **Robustness to Data Scarcity.** Perplexity on WikiText-2 with varying calibration set sizes.

| Method | Calibration Samples ($|\mathcal{D}_{\text{cal}}|$) | | | | |
|---|---|---|---|---|---|
| | 8 | 16 | 32 | 64 | 128 |
| SparseGPT | 13.27 | 13.26 | 13.25 | 13.25 | 13.25 |
| Taylor | 13.30 | 13.33 | 13.30 | 13.35 | 13.94 |
| GHOST | 14.28 | 14.26 | 14.23 | 14.23 | 14.23 |

As shown in Table 6, the joint measure (GHOST) consistently scores best and is most stable.

*Table 6.* **Saliency Ablation.** Perplexity comparison of GHOST on WikiText-2 across sparsity levels while changing the definition of $S_n^{(g)}(t)$ between $P_{k,n,p}^{(g)}(t)$, $Q_n^{(g)}(t)$, and $P_{k,n,p}^{(g)}(t) \cdot Q_n^{(g)}(t)$.

| Method | Target Sparsity | | | | |
|---|---|---|---|---|---|
| | 10% | 30% | 50% | 70% | 90% |
| $P_{k,n,p}^{(g)}(t)$ | 13.26 | 13.64 | 14.56 | 16.71 | 2158 |
| $Q_n^{(g)}(t)$ | 13.37 | 14.23 | 16.16 | 19.83 | 33.12 |
| GHOST | **13.24** | **13.51** | **14.23** | **16.16** | **25.07** |

## 6. Discussion and Limitations

**When Does GHOST Fail?** GHOST relies on the assumption that controllability and observability can be estimated from forward-pass statistics. This assumption weakens in several regimes. At extreme sparsity ($> 90\%$), perplexity degrades to 25, suggesting the retained states cannot capture essential dynamics; as shown in Appendix C, Figure 8's elbow plot, this sparsity level forces the removal of highly utilized states. Similarly, small models (130M) suffer a sharper perplexity increase (29.06 vs. 25.85 `Dense`) than larger ones, likely due to lower representational redundancy.

**Computational Overhead.** While GHOST avoids backpropagation, it requires extracting hidden states during the forward pass, adding modest overhead. On `Mamba2`-1.3B without custom kernels, GHOST calibration takes 8 minutes (2 forward passes over 128 samples), compared to $<1$ minute for `Magnitude`/`Random` and 8 minutes for `Taylor` (which additionally requires $>45$ GB VRAM).

**Generalization to Other Architectures.** GHOST is designed for `Mamba2`'s specific structure: scalar-identity $\boldsymbol{A}$ and grouped dynamics. Extending to other SSM variants (S4, H3, `Mamba1`) may require adapting the Gramian approximations. The core principle of jointly measuring input-to-state and state-to-output energy should transfer, but the specific instantiation depends on architectural details.

**Downstream Task Performance and Exact Recall.** Compressing GHOST's recurrent state limits exact-match retrieval, leading to a 14-point Lambda drop (Appendix C, Table 13). While necessary for a $\sim 20\%$ wall-clock speedup at 50% sparsity, this pruning forces a strict trade-off: tasks requiring precise factual recall degrade significantly, even as aggregate zero-shot performance stays within 2.5 points.

## 7. Conclusion

In this work, we addressed the critical inference bottlenecks of large-scale `Mamba2` models by targeting the recurrent state dimension $N$ for structured compression. Crucially, while pruning model dimension reduces the static parameter footprint, GHOST specifically accelerates sequential decoding by compressing the internal state. By bridging deep learning with control theory, GHOST approximates balanced truncation to select state channels based on their dynamic energy transfer.

Our method overcomes the "blind spots" of static magnitude metrics and the prohibitive costs of backpropagation, effectively distinguishing truly salient dynamics from misleading weight magnitudes, i.e. corporeal and phantom states. Extensive experiments confirm GHOST achieves state-of-the-art sparsity-accuracy trade-offs, rivaling expensive second-order methods without requiring backpropagation computation or massive GPU memory. GHOST not only democratizes `Mamba2` deployment by reducing memory bandwidth consumption but also demonstrates remarkable robustness to distribution shifts. Future work may extend these principles to other architectures or pair GHOST with orthogonal model compression techniques for maximum deployment efficiency.

## Impact Statement

This paper presents work whose goal is to advance the field of Machine Learning. There are many potential societal consequences of our work, none which we feel must be specifically highlighted here.

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

# A. Preliminaries

We provide expanded background on state space models and the `Mamba` architecture.

**State Space Models.** State space models describe dynamical systems via a hidden state that evolves over time. In continuous time, a linear time-invariant (LTI) system takes the form:

$$\dot{\boldsymbol{h}}(t) = \boldsymbol{A}\boldsymbol{h}(t) + \boldsymbol{B}x(t), \tag{5}$$

$$y(t) = \boldsymbol{C}\boldsymbol{h}(t) + Dx(t), \tag{6}$$

where $\boldsymbol{h}(t) \in \mathbb{R}^N$ is the hidden state, $x(t) \in \mathbb{R}$ is the input, $y(t) \in \mathbb{R}$ is the output, $\boldsymbol{A} \in \mathbb{R}^{N \times N}$ governs state dynamics, $\boldsymbol{B} \in \mathbb{R}^{N \times 1}$ maps inputs to states, $\boldsymbol{C} \in \mathbb{R}^{1 \times N}$ maps states to outputs, and $D \in \mathbb{R}$ is a direct feedthrough term.

For sequences, time is discretized using step size $\Delta \in \mathbb{R}$ converting Equations (5) and (6) into

$$\boldsymbol{h}_t = \overline{\boldsymbol{A}}\boldsymbol{h}_{t-1} + \overline{\boldsymbol{B}}x_t, \tag{7}$$

$$y_t = \boldsymbol{C}\boldsymbol{h}_t + Dx_t, \tag{8}$$

where $\overline{\boldsymbol{A}} = \exp(\Delta\boldsymbol{A})$ represents the Zero-Order Hold (ZOH) discretization, and $\overline{\boldsymbol{B}} \approx \Delta\boldsymbol{B}$ is the Euler discretization often used for efficiency.

The recurrence in Equations (7) and (8) can be viewed as a linear RNN, or equivalently unrolled into a global convolution for parallel training. This dual view underlies the efficiency of SSM-based architectures: recurrent for inference, convolutional for training.

**The `Mamba` Architecture.** `Mamba` (Gu & Dao, 2024) extends classical SSMs with selective state spaces, where the dynamics parameters $\boldsymbol{B}_t, \boldsymbol{C}_t$, and discretization step are input-dependent rather than fixed. This selectivity enables content-based reasoning: the model can modulate how strongly each input affects the hidden state and how the state influences outputs.

*Mamba2 and GQA Semantics.* `Mamba2` (Dao & Gu, 2024a) restructures the computation to leverage tensor cores, achieving significant speedups over `Mamba1`. A key architectural change is the adoption of Grouped Query Attention (GQA) semantics: the $H$ attention heads are partitioned into $G$ groups, with each group of $K = H/G$ heads sharing the same dynamics parameters $\boldsymbol{B}_t$ and $\boldsymbol{C}_t$. This sharing reduces parameter count and enables more efficient computation, but creates structural constraints for pruning. Now, removing a state channel affects all heads within a group.

*Mamba2 Block Structure.* Let $M$ be the model dimension. So, the expanded dimension $R = E \cdot M$ where $E$ is usually 2, and $R = H \cdot P$ where $H$ is the number of heads and $P$ is the head dimension. Let $N$ be the state dimension. Given layer $j$ input $\boldsymbol{u}_t^{(j)} \in \mathbb{R}^M$, a `Mamba2` block proceeds as follows.

Using pre-layer normalization, $\boldsymbol{u}_t' = \text{RMSNorm}(\boldsymbol{u}_t)$ where for an input vector $\boldsymbol{v} \in \mathbb{R}^V$, the Root Mean Square Normalization is defined as:

$$\text{RMSNorm}(\boldsymbol{v}) = \frac{\boldsymbol{v}}{\sqrt{\frac{1}{V}\sum_{i=1}^{V}\boldsymbol{v}_i^2 + \epsilon}} \odot \boldsymbol{\gamma},$$

with $\boldsymbol{\gamma} \in \mathbb{R}^V$ being a learnable scale parameter and $\epsilon > 0$ a small constant for numerical stability.

A single linear projection generates all required quantities:

$$[\boldsymbol{z}_t; \boldsymbol{x}_t; \boldsymbol{B}_t; \boldsymbol{C}_t; \boldsymbol{\Delta}_t] = \boldsymbol{W}_{\text{in}}\boldsymbol{u}_t' + \boldsymbol{b}_{\text{in}}, \tag{9}$$

where $\boldsymbol{z}_t, \boldsymbol{x}_t \in \mathbb{R}^{H \times P}$ are the gate and input signals (unique per head), $\boldsymbol{B}_t, \boldsymbol{C}_t \in \mathbb{R}^{G \times N}$ are the dynamics matrices (shared per group), and $\boldsymbol{\Delta}_t \in \mathbb{R}^H$ contains the discretization steps (unique per head).[2]

The input and dynamics undergo depthwise convolution and SiLU activation, where $\text{SiLU}(x) = x \cdot \sigma(x) = \frac{x}{1+e^{-x}}$:

$$[\boldsymbol{x}_t'; \boldsymbol{B}_t'; \boldsymbol{C}_t'] = \text{SiLU}(\text{Conv1D}([\boldsymbol{x}_t; \boldsymbol{B}_t; \boldsymbol{C}_t])). \tag{10}$$

---

[2]We follow the notation used in the literature, but we acknowledge that there is ambiguity about what is a vector (usually boldface, lowercase letters) and we have to make exceptions like $\boldsymbol{\Delta}_t \in \mathbb{R}^H$. A complete notation table is provided in Appendix B.

Next, to ensure positivity, $\boldsymbol{\Delta}'_{t,h} = \text{Softplus}(\boldsymbol{\Delta}_{t,h}) = \ln(1 + e^{\boldsymbol{\Delta}_{t,h}})$. The discretized recurrence then computes, for each head $h$ in group $g_h$:

$$\overline{\boldsymbol{A}}_{t,h} = \exp(-\boldsymbol{\Delta}'_{t,h} \cdot \boldsymbol{A}_h), \tag{11}$$

$$\overline{\boldsymbol{B}}_{t,h} = \boldsymbol{\Delta}'_{t,h} \cdot \boldsymbol{B}'_{t,g_h}, \tag{12}$$

$$\boldsymbol{H}_{t,h} = \overline{\boldsymbol{A}}_{t,h} \boldsymbol{H}_{t-1,h} + \overline{\boldsymbol{B}}^{\top}_{t,h} \boldsymbol{x}'_{t,h}, \tag{13}$$

$$\boldsymbol{y}^{\text{SSM}}_{t,h} = \boldsymbol{C}'_{t,g_h} \boldsymbol{H}_{t,h} + \boldsymbol{D}_h \boldsymbol{x}'_{t,h}, \tag{14}$$

where $\boldsymbol{A}_h \in \mathbb{R}^+$ is the learned decay for head $h$, $\boldsymbol{H}_{t,h} \in \mathbb{R}^{N \times P}$ is the hidden state, and $\boldsymbol{D}_h$ is the feedthrough scalar.

Finally, `Mamba2` gates $\boldsymbol{y}^{\text{gated}}_t = \text{RMSNorm}(\boldsymbol{y}^{\text{SSM}}_t \odot \text{SiLU}(\boldsymbol{z}_t))$, projects $\boldsymbol{y}_t = \boldsymbol{W}_{\text{out}}\text{vec}(\boldsymbol{y}^{\text{gated}}_t) + \boldsymbol{b}_{\text{out}}$ and adds a residual connection to yield the layer output $\boldsymbol{u}^{(j+1)}_t = \boldsymbol{y}_t + \boldsymbol{u}^{(j)}_t$.

*The Inference Bottleneck.* The hidden state $\boldsymbol{H}_t$ has shape $(H, N, P)$ per token. For `Mamba2`-1.3B with $H = 64$, $N = 128$, $P = 64$, this is $64 \times 128 \times 64 \times 4 = 2.1$ MB per layer in float32, or approximately 100 MB across 48 layers. During autoregressive generation, this state must be loaded and stored at every step, creating a memory-bandwidth bottleneck that structured pruning of $N$ directly addresses.

## B. Notation

Table 7 summarizes the mathematical notation used throughout this paper.

*Table 7.* **Summary of Notation.**

| Symbol | Description |
|---|---|
| *Model Dimensions* | |
| $M$ | Model dimension |
| $R$ | Expanded dimension ($R = E \cdot M = H \cdot P$) |
| $E$ | Expansion factor (typically 2) |
| $H$ | Number of attention heads |
| $P$ | Head dimension |
| $G$ | Number of groups |
| $K$ | Heads per group ($K = H/G$) |
| $N$ | State dimension |
| $L$ | Sequence length |
| *Inputs and Outputs* | |
| $\boldsymbol{u}_t \in \mathbb{R}^M$ | Layer input at time $t$ |
| $\boldsymbol{x}_t, \boldsymbol{x}'_t \in \mathbb{R}^{H \times P}$ | Input signal (before/after conv and activation) |
| $\boldsymbol{z}_t \in \mathbb{R}^{H \times P}$ | Gate signal |
| $\boldsymbol{y}_t \in \mathbb{R}^M$ | Layer output |
| $\boldsymbol{y}_t^{\text{SSM}} \in \mathbb{R}^{H \times P}$ | SSM output (before gating) |
| *SSM Dynamics* | |
| $\boldsymbol{B}_t, \boldsymbol{B}'_t \in \mathbb{R}^{G \times N}$ | Input-dependent dynamics (before/after activation) |
| $\boldsymbol{C}_t, \boldsymbol{C}'_t \in \mathbb{R}^{G \times N}$ | Input-dependent dynamics (before/after activation) |
| $\boldsymbol{\Delta}_t, \boldsymbol{\Delta}'_t \in \mathbb{R}^H$ | Discretization step sizes (before/after Softplus) |
| $\overline{\boldsymbol{A}}_{t,h} \in (0,1), \overline{\boldsymbol{B}}_{t,h} \in \mathbb{R}^{1 \times N}$ | Discretized dynamics matrices |
| $\boldsymbol{H}_{t,h} \in \mathbb{R}^{N \times P}$ | Hidden state (per head) |
| $\boldsymbol{D}_h \in \mathbb{R}$ | Feedthrough scalar for head $h$ |
| *Projections and Weights* | |
| $\boldsymbol{W}_{\text{in}} \in \mathbb{R}^{(2 \cdot H \cdot P + 2 \cdot G \cdot N + H) \times M}, \boldsymbol{b}_{\text{in}} \in \mathbb{R}^{2 \cdot H \cdot P + 2 \cdot G \cdot N + H}$ | Input projection weights and biases |
| $\boldsymbol{W}_{\text{out}} \in \mathbb{R}^{M \times H \cdot P}, \boldsymbol{b}_{\text{out}} \in \mathbb{R}^M$ | Output projection weights and biases |
| $\boldsymbol{W}_B \in \mathbb{R}^{G \cdot N \times M}, \boldsymbol{W}_C \in \mathbb{R}^{G \cdot N \times M}$ | Projection weights for $\boldsymbol{B}$ and $\boldsymbol{C}$ |
| *Gramians and Saliency* | |
| $\boldsymbol{P}_{k,n,p}^{(g)}(t)$ | Controllability of state $n$ (group $g$, head $k$, chan $p$) |
| $\boldsymbol{Q}_n^{(g)}(t)$ | Observability of state $n$ (group $g$) |
| $\boldsymbol{S}_n^{(g)}$ | Saliency score for state $n$ (group $g$) |
| $\mathcal{S}_{\text{pool}}$ | Pooled saliency scores across groups |
| *Pruning* | |
| $\boldsymbol{M} \in \{0,1\}^{N_{\text{layers}} \times G \cdot N}$ | Binary pruning mask |
| $\mathcal{D}_{\text{cal}}$ | Calibration dataset |
| $\kappa$ | Target sparsity ratio |
| $\tau$ | Pruning threshold |
| *Operations* | |
| $\| \cdot \|_F$ | Frobenius norm |
| $\| \cdot \|_0$ | $\ell_0$ pseudo-norm (number of nonzeros) |
| $\odot$ | Hadamard product |
| $\sigma(\cdot)$ | Sigmoid function $\frac{1}{1+e^{-x}}$ |
| $\text{SiLU}(x)$ | $x \cdot \sigma(x) = \frac{x}{1+e^{-x}}$ |
| $\text{Softplus}(x)$ | $\ln(1 + e^x)$ |

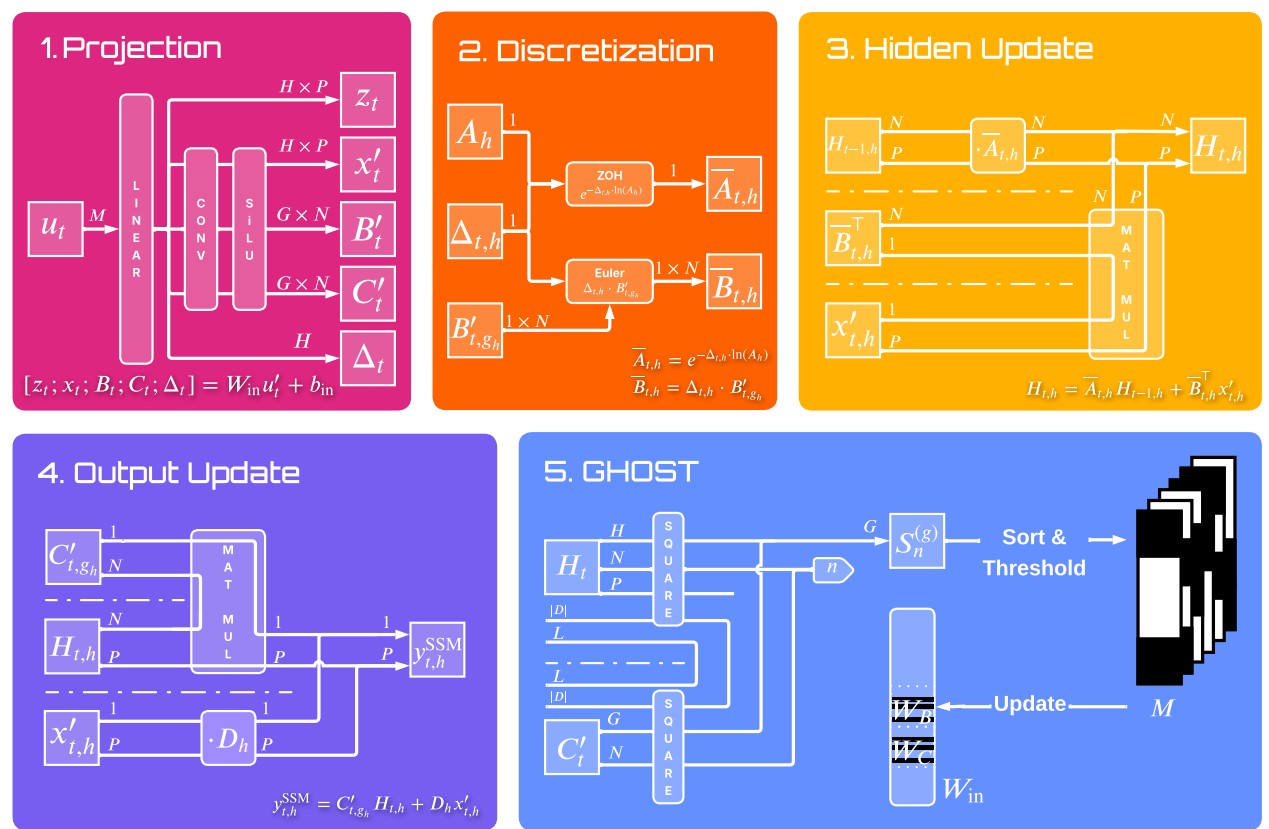

*Figure 6.* Overview of the `Mamba2` forward pass with GHOST. The input $u'_t$ is projected into intermediate variables (Subfigure 1.) and discretized parameters (Subfigure 2.) to update the hidden state $\boldsymbol{H}_{t,h}$ (Subfigure 3.) and compute the final SSM output $\boldsymbol{y}^{\text{SSM}}_{t,h}$ (Subfigure 4.). Concurrently, the GHOST mechanism computes scores $\boldsymbol{S}^{(g)}_t$ and applies sorting and thresholding to generate a binary mask $\boldsymbol{M} \in \mathbb{R}^{G \times N}$ that induces sparsity in the initial projection (Subfigure 5.).

## C. Extra Data.

*Table 8.* Impact of observability (Hessian summation) horizon on GHOST pruning performance. The marginal difference in perplexity on WikiText-2 ($\Delta$PPL $< 0.01$) suggests that immediate sensitivity is the dominant factor in saliency for `Mamba2`, justifying the computationally efficient first term choice.

| Metric | Time Horizon | | | | | |
| --- | --- | --- | --- | --- | --- | --- |
| | **1** | **4** | **16** | **64** | **256** | **1024** |
| Perplexity | 14.232 | 14.225 | 14.230 | 14.231 | 14.231 | 14.231 |

*Table 9.* **Inference Speedup.** Latency comparison (seconds) across model sizes at 50% sparsity. GHOST consistently achieves ∼20% speedup, whereas SparseGPT fails to provide meaningful hardware acceleration due to its preservation of original matrix shapes.

| Method | 130M | 370M | 780M | 1.3B | 2.7B |
|---|---|---|---|---|---|
| Dense | 11.91 | 30.23 | 43.95 | 58.37 | 96.62 |
| SparseGPT | 11.87 | 29.82 | 43.89 | 57.51 | 95.82 |
| GHOST | **9.51** | **23.92** | **35.60** | **46.13** | **72.47** |

*Table 10.* **Aggregate Wasserstein Shift for `GHOST` vs. `Taylor`.** Comparison of activation distribution shifts (measured via Wasserstein distance to the `Dense` baseline) between GHOST and Taylor pruning across varying sparsity levels.

| Method | Target Sparsity | | | | |
|---|---|---|---|---|---|
| | 10% | 30% | 50% | 70% | 90% |
| Taylor | 0.383 | 0.386 | 0.395 | 1.347 | 1.561 |
| GHOST | 0.383 | 0.383 | 0.388 | 0.397 | 0.517 |

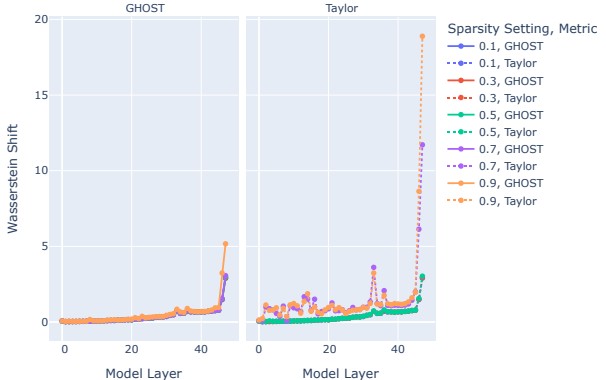

*Figure 7.* **Layerwise Wasserstein Shift for `GHOST` vs. `Taylor`.** A layerwise analysis demonstrating that `Taylor`'s instability stems from cascading distribution shifts across varying sparsity settings. Lacking the incremental calibration utilized in GHOST, significant activation distribution shifts in `Taylor`'s early layers cause downstream layers to deviate wildly, ultimately leading to the observed model collapse.

*Table 11.* **Length Robustness.** Comparison of perplexity degradation when evaluating on increasingly shorter contexts. We denote calibration free methods with *.

| Method | Calibration and Evaluation Length ($L$) | | | | | | | |
|---|---|---|---|---|---|---|---|---|
| | 16 | 32 | 64 | 128 | 256 | 512 | 1024 | 2048 |
| Dense* | 91.24 | 49.81 | 31.73 | 23.04 | 18.52 | 15.88 | 14.20 | 13.17 |
| SparseGPT | 91.37 | 49.87 | 31.79 | 23.11 | 18.58 | 15.95 | 14.27 | 13.25 |
| Magnitude* | 113.7 | 65.29 | 43.22 | 32.76 | 27.96 | 26.38 | 27.32 | 29.34 |
| Random* | 101.9 | 57.48 | 37.54 | 27.81 | 22.89 | 20.17 | 18.59 | 17.77 |
| Taylor | 139.3 | 178.9 | 276.7 | 317.3 | 499.0 | 766.3 | **14.94** | **13.94** |
| GHOST | **92.75** | **51.01** | **32.76** | **23.99** | **19.44** | **16.82** | 15.19 | 14.23 |

*Table 12.* **Gradient Distribution Metrics Across Sequence Lengths.** Note `Taylor`'s critical transition at Seq Len 512 (Table 11). Despite a low Wasserstein shift at 512 relative to 1024, indicating global distribution similarity to the 1024 baseline, the sharp drop in Average Kurtosis ($24.64 \rightarrow 15.71$) signifies "signal smearing." This flattening of reliable peaks deprives gradient-based importance metrics of their structural anchors, leading to suboptimal pruning compared to the high Gradient Signal to Noise Ratio (GSNR) short-sequence regime or the high-kurtosis long-sequence regime.

| Seq Len | Wasserstein Shift (vs 1024) | Avg GSNR | Avg Kurtosis |
|---------|-----------------------------|----------|--------------|
| 16 | 0.840 | 0.0122 | 25.11 |
| 32 | 0.390 | 0.0077 | 6.53 |
| 64 | 0.180 | 0.0049 | 19.24 |
| 128 | 0.080 | 0.0029 | 7.32 |
| 256 | 0.033 | 0.0016 | 10.54 |
| 512 | 0.011 | 0.0007 | 15.71 |
| 1024 | 0.000 | 0.0002 | 24.64 |

*Table 13.* **Zero-Shot Performance.** Accuracy on Lambada, PIQA, ARC-e, and ARC-c.

| Method | Lamb. | PIQA | ARC-e | ARC-c |
|--------|-------|------|-------|-------|
| Dense | 65.55 | 73.34 | 64.23 | 29.95 |
| SparseGPT | 62.76 | 73.29 | 64.23 | 30.12 |
| Magnitude | 17.50 | 70.08 | 58.04 | 25.68 |
| Random | 31.21 | 70.89 | 59.55 | 28.07 |
| Taylor | **59.23** | **73.39** | **64.27** | **29.86** |
| GHOST | 51.76 | 72.74 | **64.27** | 29.35 |

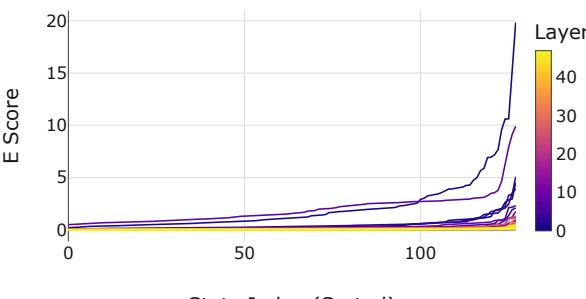

*Figure 8.* **GHOST Energy Elbow.** An elbow plot of the ghost salience score on `Mamba2`-1.3B. As pictured, the elbow is around state index $110/128 \approx 86\%$ which is roughly when perplexity on the sparsity sweep, Table 1, starts increasing too.

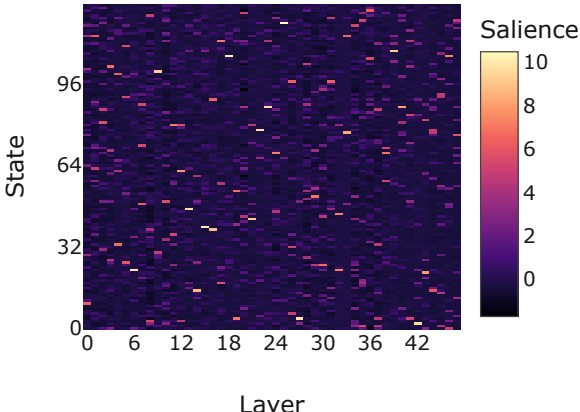

*Figure 9.* **Energy-States Heatmap.** A heat map of `GHOST` salience score for each layer by state normalized by layer.

*Table 14.* **Architectural Robustness.** Evaluation of pruning methodologies on `Zamba2` hybrid architectures. We compare the perplexity of `Zamba2`-1.2B and `Zamba2`-2.7B across varying sparsity levels. In stark contrast to the pure `Mamba2` baseline in Table 3, `Taylor`-based pruning exhibits catastrophic failure on `Zamba2`, yielding random-guess perplexity ($> 3e8$). We hypothesize this is driven by the interleaved shared attention blocks, which introduce unique structural vulnerabilities: 1) distortion of the gradient landscape which renders gradient-magnitude proxies like `Taylor` unreliable, and 2) the shared attention may inhibit adaptation to the activation distribution shifts caused by pruning the `Mamba2` backbone. Conversely, `GHOST` demonstrates resilience to this hybrid structure, maintaining perplexity near the `Dense` baseline.

| Method | 1.2B | 2.7B |
|---|---|---|
| Dense | 11.15 | 19.79 |
| SparseGPT | 11.21 | 19.76 |
| Magnitude | 1.0E5 | 51.89 |
| Random | 451.7 | 51.46 |
| Taylor | 3.0E8 | 3.5E5 |
| GHOST | **11.80** | **21.23** |

