# OpenReview forum: "GHOST: Unmasking Phantom States in Mamba2 via Grouped Hidden-state Output-aware Selection & Truncation"
_ICML.cc/2026/Conference — ICML 2026 regular_

### Official Review · Reviewer_641Y · 2026-03-09

**Soundness:** 3
**Presentation:** 3
**Significance:** 2
**Originality:** 3
**Overall Recommendation:** 4
**Confidence:** 3

**Summary:**

This paper proposes a structured pruning method for Mamba2 that targets the SSM increased hidden state dimension by identifying and removing channels that are "dynamically" less active. Inspired by control theory, it assigns each state channel a saliency score proportional to the product of a controllability proxy (capturing how strongly inputs drive the channel) and an observability (capturing how much the channel influences outputs). The proposed method only requires 2x forward-pass on a small calibration set. Experimentally, they show this is a practical gradient-free structured pruning method for Mamba2 that has smooth degradation and avoids the instability seen in Taylor structured pruning.

**Compliance With Llm Reviewing Policy:**

Affirmed.

**Final Justification:**

The rebuttal has been detailed, and it has addressed most of my comments; therefore, I update my score to 4 weak accept.

**Key Questions For Authors:**

- A point that seems unclear is the behavior of the Taylor baseline in Fig.~4: perplexity increases dramatically for intermediate short sequence lengths. Might be some flaw in the experiment?
- The number of calibration samples seems to minimally affect the method. Could the authors comment on that?

**Limitations:**

yes

**Strengths And Weaknesses:**

** Strengths

- The paper is sound and well-motivated. It tackles a current interest and open problem, as it is efficiency in SSMs by proposing a gradient-free structured pruning to reduce state dimension.
- The method is well grounded in control theory, and practically implemented through the defined controllability and observability proxies.
- The method seems to perform comparably to Taylor but without the need for backpropagation, instead a 2 times forward-prop.

** Weaknesses

-  While SparseGPT primarily sparsifies weights in linear layers (i miss more information in the experimentation conditions), GHOST prunes the SSM state dimension. The paper would benefit from further clarification on the fairness of the comparison and a more details on the evaluation would be needed.
- The paper would benefit of a discussion on when one should prune the SSM internal state channels versus the outher linear layers. Other recent works (e.g. biMamba) and post-training quantization techniques point that the majority of the parameters to tackle are in the linear layers which might represent over ~90 percent of the parameters of the model. Seems SparseGPT is also in that direction, and more thorough comparison would benefit the work.

---

> ### Author Rebuttal · Authors · 2026-03-30
>
> We are grateful for the reviewer’s recognition of GHOST as a sound and well-motivated approach to the open problem of SSM efficiency. It is rewarding that both our control-theoretic grounding and the practical efficiency of our backprop-free proxies resonated, particularly as our method achieves performance parity with Taylor-based pruning while requiring only two forward passes.
>
> # Structured vs. Unstructured Fairness
>
> To clarify "fairness": we apply GHOST and SparseGPT to the exact same matrices ($\boldsymbol{W} _{\boldsymbol{B}}$ and $\boldsymbol{W} _{\boldsymbol{C}}$). The core difference is the **granularity**: SparseGPT removes n% of individual elements, while GHOST removes n% of entire rows. We chose this comparison because of the scarcity of open source structured pruning methods for Mamba2. Please see our response to reviewer PznT, for a highlight of how GHOST achieves strong performance while enabling wall clock speedups that SparseGPT cannot provide.
>
> # State Pruning vs. Outer Linear Layer Pruning
>
> We thank the reviewer for giving us the opportunity to clarify this architectural distinction. You note a discussion on "when one should prune the SSM internal state channels versus the outher linear layers". To clarify, GHOST _does_ prune linear weights. Specifically, the $\boldsymbol{W} _{\boldsymbol{B}}$ and $\boldsymbol{W} _{\boldsymbol{C}}$ slices of $\boldsymbol{W} _{\text{in}}$ that project into and out of the SSM state dimension.
>
> We suspect the reviewer is referring to pruning the primary channel expansion projections (which sends vectors of size $D \rightarrow ED$), which indeed house the vast majority of the static parameters. We view pruning the state dimension and pruning the outer channel dimension as solving two completely different hardware bottlenecks. Pruning the outer linear layers reduces the _static parameter footprint_. Conversely, pruning the SSM internal state specifically targets the _dynamic recurrent memory bottleneck_ (analogous to compressing a Transformer's KV cache) and accelerates the sequential decoding step.
>
> Because they target different dimensions, GHOST is completely orthogonal to outer-layer pruning or post-training quantization. We will add a discussion to the final paper clarifying that for maximum deployment efficiency, GHOST should be paired with existing outer-layer compression techniques.
>
> # The Taylor "Hump" at 512 Seq Len
>
> Our investigations reveal that the spike in Taylor’s perplexity at intermediate sequence lengths (512) is not a flaw, but a reflection of "signal smearing." At 512, the average kurtosis of the gradient distribution drops sharply (24.64 $\rightarrow$ 15.71) compared to 1024. While the distribution still looks similar to the 1024 setting (low Wasserstein shift), the "reliable peaks" in the gradient signal flatten out. Taylor loses its "anchors" for importance, leading to suboptimal pruning choices that don't occur in the more "peaky" (high kurtosis) long-sequence or low-noise (high gradient signal to noise ratio, GSNR) short-sequence regimes.
>
> | Seq Len | Wasserstein Shift (vs 1024) | Avg GSNR | Avg Kurtosis |
> | ------- | --------------------------- | -------- | ------------ |
> | 16      | 0.84                        | 0.0122   | 5.11         |
> | 32      | 0.39                        | 0.0077   | 6.53         |
> | 64      | 0.18                        | 0.0049   | 19.24        |
> | 128     | 0.08                        | 0.0029   | 7.32         |
> | 256     | 0.033                       | 0.0016   | 10.54        |
> | 512     | 0.011                       | 0.0007   | 15.71        |
> | 1024    | 0                           | 0.0002   | 24.64        |
> # Calibration Sample Invariance
>
> The observation that calibration sample count minimally affects performance is consistent with recent findings in pruning literature: "We observe that the average performance of pruned models is robust to data amount, regardless of the calibration data source, with fluctuations of only 0.1%-0.2%" (1).
>
> ## References
>
> (1) https://arxiv.org/pdf/2410.17711

---

> > ### Author Rebuttal · Reviewer_641Y · 2026-04-04
> >
> > I thank the authors for their response, which addresses my concerns. I will consider the evaluation of the paper accordingly.

---

> > > ### Author Response · Authors · 2026-04-04
> > >
> > > Thank you for the positive feedback and for the time spent evaluating our rebuttal. We are glad to hear that our clarifications regarding the architectural distinctions of GHOST and the analysis of the Taylor baseline fully addressed your concerns. We look forward to incorporating these discussions and the additional data into the final version of the manuscript to improve its clarity and impact.

---

### Official Review · Reviewer_k89k · 2026-03-11

**Soundness:** 3
**Presentation:** 2
**Significance:** 3
**Originality:** 3
**Overall Recommendation:** 5
**Confidence:** 3

**Summary:**

GHOST is a structured pruning method targeting Mamba2's recurrent hidden state dimension N. The core observation is that standard magnitude pruning fails for Mamba2 because weight magnitude does not correlate with runtime importance. Some channels with small weights are highly active at inference time ("phantom states") while some large-weight channels contribute little ("corporeal states"). To address this, GHOST computes a saliency score combining controllability (how strongly inputs drive the state, via hidden-state covariance) and observability (how much the state affects outputs, via an instantaneous Hessian proxy). The scoring respects Mamba2's GQA structure, pools across groups, and performs layer-by-layer sequential pruning with activation updates. The method requires only two forward passes over calibration data, with no gradients and approximately 15GB VRAM for a 1.3B model.

**Compliance With Llm Reviewing Policy:**

Affirmed.

**Key Questions For Authors:**

1. Can you quantify phantom and corporeal state prevalence? For example: what fraction of channels per layer have high dynamic energy but low weight norm, and vice versa? This would help calibrate how severe the magnitude-pruning failure actually is.

2. The instantaneous observability proxy drops all future time steps. Have you tried a short-horizon variant - say, summing the Hessian over the next 5-10 steps? This should be computationally inexpensive and might help preserve longer-range dependencies.

3. SparseGPT is competitive across the board in the main table and outperforms GHOST at low sparsity. Since it was designed for transformers, do you have an explanation for why it transfers well to Mamba2?

**Limitations:**

yes

**Strengths And Weaknesses:**

Strengths

The phantom state finding is the key contribution. I have not seen this failure mode clearly documented for Mamba2 before, and it illustrates why transformer pruning heuristics do not transfer to SSMs. The scatter plot showing weight magnitude versus dynamic energy communicates this point well.

The control-theoretic framing is appropriate. Mamba2's recurrence fits naturally into the LTV system perspective, and balanced truncation gives the saliency score a principled basis rather than relying on another heuristic. The derivation of the instantaneous observability approximation is clean (justified by exponential decay in discretized transitions), and connecting the saliency score to expected reconstruction error provides useful theoretical grounding.

Experimental coverage is thorough. The sparsity sweep shows a clear pattern: GHOST and Taylor are comparable at low sparsity, but GHOST degrades more gracefully at higher sparsity while Taylor's performance collapses. The sequence-length robustness experiments are informative as well - Taylor's perplexity increases sharply at longer evaluation lengths. The scaling experiments across 130M-2.7B are useful, particularly the finding that Taylor fails on smaller models.

Weaknesses

Notation and exposition need improvement. Many symbols are introduced in quick succession in the preliminaries, and I had to reread the saliency scoring section to follow how the controllability and observability estimates are computed in practice. It would help to explain what each term captures at an intuitive level before presenting the formal definitions. The overview figure is informative but dense - a simpler diagram focused on the scoring mechanism would improve clarity.

The phantom state finding may be overstated. Looking at the scatter plot more carefully, the phantom cases (low weight, high energy) appear to be a minority - most points sit near the origin or along the diagonal. The paper does not quantify what fraction of states are phantoms versus corporeal. This matters: if phantoms are rare, magnitude pruning may not encounter them often, and the failure mode could be less severe than suggested.

The instantaneous observability approximation is the largest theoretical shortcut, and the justification beyond "exponential decay" is limited. There could be state channels with weak moment-to-moment observability but significant cumulative influence over longer sequences - for example, channels tracking slowly-varying context. Comparing against a multi-step Hessian estimate, even on a small model, would help quantify how much this approximation costs.

On downstream tasks: the zero-shot numbers are weaker than the perplexity improvements suggest. GHOST achieves about 59% on Lambada versus 75% for the dense model, which is a large gap. The paper focuses on outperforming Magnitude/Random baselines, which is not a high bar; for practical deployment, the gap to the unpruned model is more relevant.

---

> ### Author Rebuttal · Authors · 2026-03-30
>
> We appreciate the reviewer’s recognition of the "phantom state" as a key contribution that explains why traditional transformer pruning heuristics fail when applied to SSMs. Furthermore, we are encouraged by the positive feedback on our principled control-theoretic framing and the demonstrated robustness of our method across varying sequence lengths and model scales.
>
> # Improving Exposition
>
> Thank you for sharing how our exposition can be hard to follow. As a result, we have outlined several improvements to notation and language, which we outline in our response to Reviewer PznT.
>
> We also thank you for the practical guidance on improving our main figure. We are moving the current figure to the appendix for readers who want to deep dive into the math of Mamba2 and will keep a simplified version focusing primarily on box 5 (the selection mechanism) of the original figure.
>
> # Quantifying Phantom and Corporeal States
>
> To calibrate the severity of the magnitude-pruning failure, we calculated the prevalence of these states in Figure 1. Using a 50% sparsity threshold, we found:
>
> - **Phantom States** (High energy/Low weight): 20.66%
>
> - **Corporeal States** (Low energy/High weight): 20.64%
>
>
> Total misclassification by weight-based methods is **41.3%**, with a Pearson Correlation of **-0.1940**. This confirms that weight magnitude is a poor proxy for SSMs.
>
> # Instantaneous Observability & Time Horizons
>
> Regarding the "instantaneous" approximation: we tested varying the Hessian summation horizon ($T$) to see if capturing longer-range dependencies improved results.
>
> | **Time Horizon (T)** | **1**  | **4**  | **16** | 64     | 256    | **1024** |
> | -------------------- | ------ | ------ | ------ | ------ | ------ | -------- |
> | **Perplexity**       | 14.232 | 14.225 | 14.230 | 14.231 | 14.231 | 14.231   |
>
> The difference is marginal ($<0.1$ PPL). This suggests that for Mamba2, the immediate sensitivity of the output to the state is the dominant factor in saliency, justifying our computationally efficient $T=1$ choice.
>
> # Lambada Performance
>
> We appreciate the reviewer's critical eye regarding the gap to the unpruned model on downstream tasks, which is indeed the most relevant metric for practical deployment.
>
> Looking at Table 9, we acknowledge the 14-point discrepancy on Lambada. Mechanistically, Lambada evaluates exact last-word prediction, a task heavily reliant on precise factual recall and long-range context preservation. Because GHOST structurally compresses the recurrent hidden state, which acts as the model's working memory, it is expected that tasks requiring highly specific, exact-match token retrieval will degrade more noticeably than broad semantic tasks or general perplexity.
>
> However, we emphasize that this represents a practical Pareto trade-off. Compressing this memory state is precisely what allows GHOST to achieve a **~20% wall-clock speedup** at 50% sparsity. Furthermore, the aggregate difference across all zero-shot settings remains less than 2.5 percentage points. This indicates that outside of exact-match recall extremes, the general reasoning and language modeling capabilities remain highly robust. We will add this nuanced analysis to our limitations section.
>
> # SparseGPT Performance
>
> Since $\boldsymbol{W} _{\boldsymbol{B}}$ and $\boldsymbol{W} _{\boldsymbol{C}}$ are projection matrices, SparseGPT works well as it was designed as a method to effectively sparsify projection weights by minimizing the local reconstruction error. SparseGPT does not care about the global context as it focus on preserving the input/output relationship across a single matrix transform. Additionally, being an unstructured method, one key boon to SparseGPT's performance is its ability to use the remaining elements in a row of the projection matrix to compensate for the missing contribution of the pruned elements. However, as noted in the response to Reviewer PznT, SparseGPT’s accuracy comes at the cost of speed. GHOST tackles the harder "structured" problem, i.e. deleting entire rows, which is necessary for the memory and wall-clock savings demonstrated in our results (please see our empirical findings in our response to PznT).

---

> > ### Author Rebuttal · Reviewer_k89k · 2026-04-04
> >
> > I have no further concerns, and my original score stands.

---

> > > ### Author Response · Authors · 2026-04-04
> > >
> > > Thank you for the final confirmation that all concerns are fully resolved. We are pleased that the additional quantification of phantom states and the time-horizon ablations provided the necessary clarity to support our findings. We appreciate your thoughtful engagement throughout the process and your recommendation for acceptance.

---

### Official Review · Reviewer_PznT · 2026-03-12

**Soundness:** 3
**Presentation:** 2
**Significance:** 3
**Originality:** 3
**Overall Recommendation:** 4
**Confidence:** 4

**Summary:**

This paper proposes Grouped Hidden-state Output-aware Selection and Truncation (GHOST), a method for pruning the hidden-state channels of Mamba-2 to compress the recurrent state dimension and thereby target lower inference-time memory and computational cost. The method is gradient-free and defines a data-dependent saliency score for each state channel using two factors: controllability, i.e. how much the input history has contributed to the current state, and observability, i.e. how strongly that channel affects future outputs. This score is estimated using forward-pass statistics on calibration data and is then used to order the states for pruning. The experiments evaluate GHOST across sparsity levels, sequence-lengths, model size, zero-shot transfer, calibration-data scarcity, and an ablation of the saliency score. Overall, the results suggest that GHOST’s main strength is robustness: it degrades smoothly at high sparsity and remains stable across model scales, while the ablation supports the use of the full controllability-observability score rather than either term alone.

**Compliance With Llm Reviewing Policy:**

Affirmed.

**Final Justification:**

I am raising my recommendation from weak reject to weak accept. My initial concern was that, although the paper is original, technically well motivated, and supported by broad experiments, the presentation needed improvement and the case against SparseGPT as a fair baseline was not convincing enough. The rebuttal addressed this main issue by providing empirical evidence that hard pruning with GHOST yields real runtime gains, whereas SparseGPT does not materially improve efficiency in this setting, which changed my evaluation. The rebuttal also gave a plausible explanation for Taylor’s instability and committed to clear fixes for the notation, Figure 1, and some imprecise wording. Overall, I now view the paper as sound, practically relevant, and sufficiently significant to merit acceptance.

**Key Questions For Authors:**

1) Can you provide some additional justification, in particular empirical justification, for why SparseGPT should not be considered a fair baseline here?
2) Did you run any interpretability experiments to try to understand why Taylor is so unstable, especially at high sparsity and on smaller models?

A strong answer to (1), together with improvements to the presentation of Mamba-2 and Figure 1, would likely raise my score.

**Limitations:**

Yes.

**Strengths And Weaknesses:**

# Strengths

- The proposed saliency score is well motivated from three complementary perspectives: the informal intuition, the local loss viewpoint, and the complexity analysis.
- The experiments cover a broad range of practically relevant settings, including different sparsity levels, evaluation lengths, calibration-data budgets, and model sizes.
- The proposed method demonstrates substantially better robustness than Taylor and the static pruning baselines.

# Weaknesses

- The presentation of Mamba-2 should be improved. First, it is not clear from the main body that $\odot$ denotes elementwise multiplication with broadcasting across the $P$ dimension, rather than the usual Hadamard product. Second, $\bar{A}_{t,h}$ is defined as $\exp(\Delta{t,h}\log(\mathbf{A}_h))$, but the appendix describes $A_h = a_hI$ as taking negative values so this expression is not well-defined over the reals as written. Third, Table 7 defines $A \in \mathbb{R}^{N \times N}$, making the intended type of $A_h$ unclear.
Fourth, the model is introduced using a single stacked input projection $\mathbf{W}\_{\mathrm{in}}$, but Figure 1, Section 4.5, and Table 7 later refer to separate projection weights $\mathbf{W}_B$ and $\mathbf{W}_C$ without clearly explaining their notation or relationship to $\mathbf{W}\_{\mathrm{in}}$.
- Similarly, the presentation of Figure 1 should be improved. The figure plays a foundational motivational role, but it is not sufficiently self-explanatory. In particular, it is unclear what a "state" denotes in this plot, and the caption does not explain clearly enough why the score based on $W_B$ and $W_C$ should be regarded as an appropriate static proxy for state importance. As written, the figure becomes intuitive only after the reader has already worked through the later sections of the paper, rather than serving as an effective upfront motivation.
- The introductory slogan “No gradients. No Hessians. No quadratic memory overhead.” contrasts somewhat with the otherwise academic tone of the paper. It is also technically inaccurate, since GHOST explicitly introduces a Hessian of the output energy with respect to the hidden state in deriving the observability score, although I suspect this is not the type of Hessian the slogan was intended to rule out.
- The paper does not fully convince me that SparseGPT is not a comparable baseline. In the reported results, SparseGPT consistently achieves better perplexity than GHOST, and Figure 3 reports essentially the same peak VRAM for the two methods. The case against SparseGPT therefore rests on an asymptotic argument about calibration FLOPs and the claim that unstructured pruning leaves Mamba2’s recurrent state dense. Since the paper does not provide empirical evidence that this density actually prevents practical speedups or bandwidth reductions after pruning, I do not think the exclusion of SparseGPT as a fair baseline is fully justified. Even a small-scale experiment measuring activation density or runtime after pruning would make this argument much more convincing.

---

> ### Author Rebuttal · Authors · 2026-03-30
>
> We appreciate the reviewer’s positive assessment of our three-perspective motivation and broad experimental coverage of practically relevant settings. It is encouraging that they recognized our method’s robustness compared to existing Taylor and static pruning baselines.
>
> # Clarifying Mamba2 Notation
>
> We appreciate the feedback on mathematical clarity. To ensure a more rigorous presentation of the State Space Model framework, we will:
>
> - Refine the definition of $\bar{A}$ to avoid broadcasting issues, framing it as a scalar where $\bar{A}_{t, h} = \text{exp} ( \Delta _{t, h} \log (A _{h})))$ with $A _h \in \mathbb{R}^+$.
>
> - Update **Table 7** to reflect $A \in \mathbb{R}^H$ with $A _h \in \mathbb{R}^+$ for $h \in [H]$.
>
> - Explicitly define $\boldsymbol{W} _{\boldsymbol{B}}$ and $\boldsymbol{W} _{\boldsymbol{C}}$ in the Section 2 preliminaries as the specific slices of $\boldsymbol{W} _{\text{in}}$ responsible for generating the $B$ and $C$ tensors.
>
>
> # Improving Language
>
> To clarify "states" and the score proxies, we are updating the Figure 1 caption to specify that:
>
> - Akin to an element of a Transformer's KV cache, a **state** refers to an individual coordinate in a Mamba layer's hidden state recurrent memory representation of a sequence.
>
> - In the Mamba2 recurrence, $\boldsymbol{W} _{\boldsymbol{B}}$ transforms inputs into the hidden state space and $\boldsymbol{W} _{\boldsymbol{C}}$ transforms the hidden state space into the output space. Thus, one may naively expect that using weight magnitude is a suitable proxy for state importance.
>
> Next, while our importance score is analytically grounded in the Hessian, we appreciate the feedback regarding our terminology and will remove the slogan to ensure technical precision. We will clarify that the method's efficiency stems from operating entirely on forward-pass activations, thereby eliminating the overhead of backpropagation and persistent computation graphs during pruning.
>
> # SparseGPT as an Unfair Baseline
>
> To clarify the distinction between method fairness, we categorize SparseGPT as an unstructured pruning baseline (deletes weight elements not rows). This makes it an "optimistic" comparison for accuracy but an "unfair" one for efficiency. To justify this empirically, we measured inference wall-clock time after implementing hard pruning (physical matrix contraction):
>
> | **Model** | **Dense (s)** | **SparseGPT (s)** | **GHOST (s)** |
> | --------- | ------------- | ----------------- | ------------- |
> | 130M      | 11.91         | 11.87             | **9.51**          |
> | 370M      | 30.23         | 29.82             | **23.92**         |
> | 780M      | 43.95         | 43.89             | **35.60**         |
> | 1.3B      | 58.37         | 57.51             | **46.13**     |
> | 2.7B      | 96.62         | 95.82             | **72.47**     |
>
> At 50% sparsity, GHOST is **~20% faster** than SparseGPT on average. SparseGPT yields <1% speedup over dense models because it leaves the matrix shape intact, making it unsuitable for real-world hardware acceleration in this context.
>
> # Why Taylor is Unstable
>
> At a high level, using the Wasserstein distance to measure activation distribution shifts between pruned and dense models, we observed that Taylor's PPL failure (seen in Table 1) aligns with activation distribution shits (below). Indicating that activation shifts are a good indicator for PPL collapse.
>
> | Sparsity | GHOST | Taylor  |
> | -------- | ----- | ------- |
> | 0.1      | 0.383 | 0.383   |
> | 0.3      | 0.383 | 0.386   |
> | 0.5      | 0.388 | 0.395   |
> | 0.7      | 0.397 | **1.347** |
> | 0.9      | 0.517 | **1.561** |
>
> A layerwise analysis suggests Taylor’s instability stems from **cascading** distribution shifts. Since Taylor lacks the incremental calibration used in GHOST; once an early layer's activation distribution shifts significantly, we see downstream layer's deviate wildly, leading to the observed model collapse. We will add our layerwise results as a line graph to the appendix.

---

> > ### Author Rebuttal · Reviewer_PznT · 2026-04-03
> >
> > Thank you for the detailed response, this addresses all of my main concerns, and I intend to raise my score.
> >
> > My only remaining request is for some further detail on how the “hard pruning” is implemented.

---

> > > ### Author Response · Authors · 2026-04-04
> > >
> > > Thank you for your positive feedback and for the intention to raise your score. We are glad that our previous responses addressed your main concerns.
> > >
> > > Regarding the implementation of "hard pruning," we provide the specific structural steps below. While soft pruning uses a binary mask to zero out channels, our hard pruning acts as a physical compression:
> > > 1. We initialize a new Mamba2 layer with a reduced state dimension corresponding to the number of active channels in the mask.
> > > 2. Parameters independent of the state dimension (e.g. the output projection) are copied directly to the new layer.
> > > 3. For parameters dependent on the state dimension, we use the binary mask to index/slice the original parameter tensors, creating a compressed view that is then copied into the new layer's tensors.
> > > 4. The original layer is replaced by the new, smaller layer in the model architecture and the original layer is deleted to free memory.
> > >
> > > This ensures that the model realizes actual throughput gains and memory reduction. We will include this step-by-step breakdown in Section 4.5 of the final manuscript. Thank you again for your constructive engagement with our work.

---

### Decision · Program_Chairs · 2026-04-30

**Decision:**

Accept (regular)

**Comment:**

This submission proposes a structured pruning method for Mamba-2 that aims to reduce its large hidden state dimension. Inspired by control theory, it assigns a saliency score for each channel based on a controllability and observability score. The proposed method requires only 2 forward passes and is practical, and is substantially more robust than baselines. After rebuttal, our viewers recommend acceptance.